# Potent Anticancer Activity of CXCR4-Targeted Nanostructured Toxins in Aggressive Endometrial Cancer Models

**DOI:** 10.3390/cancers15010085

**Published:** 2022-12-23

**Authors:** Esperanza Medina-Gutiérrez, Annabel García-León, Alberto Gallardo, Patricia Álamo, Lorena Alba-Castellón, Ugutz Unzueta, Antonio Villaverde, Esther Vázquez, Isolda Casanova, Ramon Mangues

**Affiliations:** 1Oncogenesis and Antitumor Drugs Group, Institut d’Investigació Biomèdica Sant Pau (IIB Sant Pau), 08041 Barcelona, Spain; 2Oncogenesis and Antitumor Drugs Group, Institut de Recerca Contra la Leucèmia Josep Carreras, 08025 Barcelona, Spain; 3Department of Pathology, Hospital de la Santa Creu i Sant Pau, 08041 Barcelona, Spain; 4CIBER de Bioingeniería, Biomateriales y Nanomedicina (CIBER-BBN), 28029 Madrid, Spain; 5Departament de Genètica i de Microbiologia, Universitat Autònoma de Barcelona, 08193 Bellaterra, Spain; 6Institut de Biotecnologia i de Biomedicina, Universitat Autònoma de Barcelona, 08193 Bellaterra, Spain

**Keywords:** advanced endometrial cancer, metastasis, protein nanoparticles, bacterial toxins, targeted drug delivery, CXCR4

## Abstract

**Simple Summary:**

The fact that most endometrial cancer (EC) patients overexpress the CXCR4 receptor in tumor tissue, especially at incurable advanced stages, opens an avenue for developing novel therapies targeting CXCR4^+^ EC cells to add new armamentarium against this malignancy. We have generated novel CXCR4-targeted nanotoxins, T22-DITOX-H6 and T22-PE24-H6, which we here evaluate in EC mouse models. We observed a selective killing of CXCR4^+^ EC cells by apoptosis induction in cultured cells as well as in tumor models, which inhibit tumor growth and increase mouse survival after repeated intravenous doses. Moreover, T22-DITOX-H6 induced a significant block of metastasis dissemination without toxicity in non-tumor tissues. Therefore, both nanotoxins may become alternative therapeutics for CXCR4^+^ high-risk EC patients, who nowadays lack effective therapies.

**Abstract:**

Patients with advanced endometrial cancer (EC) show poor outcomes. Thus, the development of new therapeutic approaches to prevent metastasis development in high-risk patients is an unmet need. CXCR4 is overexpressed in EC tumor tissue, epitomizing an unexploited therapeutic target for this malignancy. The in vitro antitumor activity of two CXCR4-targeted nanoparticles, including either the *C. diphtheriae* (T22-DITOX-H6) or *P. aeruginosa* (T22-PE24-H6) toxin, was evaluated using viability assays. Apoptotic activation was assessed by DAPI and caspase-3 and PARP cleavage in cell blocks. Both nanotoxins were repeatedly administrated to a subcutaneous EC mouse model, whereas T22-DITOX-H6 was also used in a highly metastatic EC orthotopic model. Tumor burden was assessed through bioluminescence, while metastatic foci and toxicity were studied using histological or immunohistochemical analysis. We found that both nanotoxins exerted a potent antitumor effect both in vitro and in vivo via apoptosis and extended the survival of nanotoxin-treated mice without inducing any off-target toxicity. Repeated T22-DITOX-H6 administration in the metastatic model induced a dramatic reduction in tumor burden while significantly blocking peritoneal, lung and liver metastasis without systemic toxicity. Both nanotoxins, but especially T22-DITOX-H6, represent a promising therapeutic alternative for EC patients that have a dismal prognosis and lack effective therapies.

## 1. Introduction

Endometrial cancer (EC) is the most common gynecological cancer in western countries. It involves 90% of all uterine cancers. It accounts for 90,000 deaths worldwide per year [1], whereas the five-year survival of patients diagnosed at advanced stages is only 20% [2,3]. Despite the recent increase in molecularly targeted therapies approved by the FDA, including immune checkpoint inhibitors (pembralizumab, dostarlimab), angiogenesis inhibitors (lenvatinib, bevacizumab) or mTOR inhibitors (everolimus, temsirolimus), the first line therapy remains still surgery, chemotherapy and radiotherapy [4]. The modest response to conventional treatments, along with their high rates of resistance to chemotherapy, explains why current therapies are not successful in treating high-risk, recurrent or metastatic EC patients [5].

Thus, there is an unmet medical need calling to develop new therapeutic approaches that improve current treatment, both by preventing metastatic development in high-risk EC patients and achieving a higher inhibition of metastatic dissemination, along with a significant reduction in its associated systemic toxicity and severe adverse effects [6,7]. Moreover, while selective therapeutic alternatives such as antibody–drug conjugates (ADC), consisting of a targeted monoclonal antibody chemically attached to a cytotoxic agent, have been approved for other cancer types [8]; however there are none currently available to treat EC [9].

The development of toxin-based therapies, and particularly the immunotoxins displaying antitumor activity, is a highly dynamic research field, which is being evaluated in clinical trials despite their low therapeutic window. Immunotoxins are similar to ADC since they pursue targeted drug delivery; thus, they consist of a targeting monoclonal antibody, or cytokine, chemically linked to a cytotoxic domain of bacterial, plant or animal toxins [10]. Among the clinical trials that evaluated toxin-based therapies against cancer, however, the FDA approved only those to treat hematological neoplasias. In contrast, immunotoxins did not show sufficient activity against solid tumors because of poor specificity and severe adverse effects [11].

To prevent these handicaps, we are developing novel nanoparticles displaying multivalent-CXCR4 targeting ligands. These nanoparticles are capable of selectively delivering conjugated drugs or cytotoxic protein domains to cancer cells that overexpress the CXCR4 receptor (CXCR4^+^) in cancer tissues, avoiding their delivery in healthy organs because of the low or negligible CXCR4 expression level in normal cells [12,13]. Thus, our goal is to tackle the urgent need to develop therapeutic agents that could be exploited to achieve targeted delivery of bacterial exotoxins to CXCR4^+^ tumor cells and thus reach personalized therapy that enhances the antineoplastic effect while reducing adverse effects in EC patients [14].

CXCR4 is a chemokine receptor overexpressed in EC tumor cells both at mRNA and protein levels [15], showing mostly membrane staining in cancer tissues of EC patients, which has been reported to promote metastases in our previous work [16]. We also recently proved the role of CXCR4 in enhancing metastasis in the EC mouse model. Moreover, we demonstrated a CXCR4-dependent biodistribution for the T22-GFP-H6 nanocarrier, and its enhanced uptake in tumor tissue in a subcutaneous EC model. In this work, we selected the CXCR4^+^ Luciferase^+^ AN3CA cell line, among the tested HEC1A, ARK-2 and AN3CA cells, to generate the subcutaneous and metastatic EC models to be used in the evaluation of these novel therapeutic nanoparticles because of their higher internalization of our CXCR4-targeted nanocarrier in this cell line. Moreover, the orthotopic model derived from CXCR4^+^ Luciferase^+^ AN3CA yielded the highest metastatic and aggressive malignancy [16].

In these models, two nanotoxins were tested: T22-DITOX-H6, which incorporates the deimmunized catalytic domain of the exotoxin from *Corynebacterium diphtheriae* (DITOX) and T22-PE24-H6, which incorporates the *Pseudomonas aeruginosa* (PE24) exotoxin domain as previously described [17] (Appendix A). Both nanotoxins maintained the same nanostructure as that described for the T22-GFP-H6 nanocarrier and, similarly, formed self-assembled protein-only nanoparticles that achieve a multivalent display of CXCR4-targeted T22 ligands. These nanotoxins had previously shown high preclinical therapeutic potential in head and neck and colorectal cancer models and in hematological cancer models [18,19,20]. Additional therapeutic nanoparticles derived from the same carrier that include human proapoptotic domains, specifically BAK, PUMA or BAX pore-forming domains [13], are here compared with the nanotoxins regarding their cytotoxic effect in vitro.

Thus, in this work, we performed a screening of nanotoxins and proapoptotic nanoparticles in terms of their anticancer effect in vitro, selecting only those with a higher effect, namely T22-DITOX-H6 and T22-PE24-H6. Then, we evaluated the antitumor and antimetastatic effect of the nanotoxins in aggressive subcutaneous and orthotopic EC models. Our results support the nanotoxins as alternative therapies that could improve the current treatment of high-risk EC because of their capacity to prevent metastasis development at the clinically relevant dissemination sites for EC using our recently developed orthotopic model of metastatic CXCR4^+^ EC.

## 2. Materials and Methods

### 2.1. Cell Culture

AN3CA cell line (ATCC) and transduced CXCR4^+^ Luciferase^+^ AN3CA were cultured in DMEM and F12 medium 1:1, while HEC1A (ATCC) and transduced CXCR4^+^ Luciferase^+^ HEC1A were cultured in high glucose DMEM. All cell lines were maintained in their medium supplemented with 10% fetal bovine serum and penicillin/streptomycin, at 37 °C and 5% CO_2_. The CXCR4^+^ Luciferase^+^ AN3CA cell line was obtained previously by lentiviral transduction in a previous work by our group. CXCR4 cell expression was assessed before all in vitro and in vivo experiments by flow cytometry as previously described [16].

#### 2.1.1. Cytotoxic Effect

In vitro cytotoxicity was evaluated using XTT assay (Roche, Basel, Switzerland) following the manufacturer’s instructions; 25,000 cells per well were seeded in 96 well plates and exposed to therapeutic nanoparticles at different concentrations ranging from 1 nM to 1000 nM for 48 h. A competition assay was performed by preincubating the cells with 1 µM CXCR4 antagonist AMD3100 for 1 h, followed by nanoparticle exposure. Absorbance at 490 nm was read (FLUOstar OPTIMA, BMG Labtech, Ortenberg, Germany) and cell viability was quantified and expressed in relation to the viability of buffer-treated cells.

#### 2.1.2. Apoptosis Assessment by Flow Cytometry

The percentage of cells undergoing early or late apoptosis was quantified using an Annexin V-FITC/propidium iodide (PI) detection kit (Sigma-Aldrich, St. Louis, MO, USA) following the manufacturer’s instructions. Data were analyzed using MACSQuant cytometer and MACS Quantify v2.3 software (Miltenyi Biotec, Bergisch Gladbach, Germany).

#### 2.1.3. Paraffin-Embedded Cell Blocks

In total, 3.5 × 10^6^ CXCR4^+^ Luciferase^+^ AN3CA cells were seeded in 150 cm^2^ cell culture flasks. Then, 24 h later, they were treated with either buffer or 4 nM of T22-DITOX-H6 or T22-PE24-H6 nanotoxins for 6, 12, 24 or 48 h. Supernatants were collected and attached cells were washed, trypsinized, collected in the supernatant and washed in PBS. Cell blocks were obtained by gently mixing 2 drops of human plasma and 2 drops of human thrombin (Roche). Cell blocks were fixed in paraformaldehyde 4% and embedded in paraffin for further immunocytochemistry or DAPI staining analysis. All slides were examined under an optic or fluorescence microscope (Olympus BX53, Olympus, Tokyo, Japan) and representative pictures were taken using an Olympus DP73 camera.

#### 2.1.4. DAPI Fluorescent Staining

ProLong Gold Antifade Mountant with DAPI (Invitrogen, Waltham, MA, USA) was used to assess DNA condensation and apoptotic cell bodies in paraffin slides of cell blocks. 4 µM slides were incubated at 60 °C for 1 h and rehydrated with 3 × 5 min incubations in xylene and decreasing concentrations of alcohol to 76° to be stained with DAPI.

#### 2.1.5. Immunocytochemistry

Anti-active caspase-3 (1:300; BD Pharmingen, 559565) and anti-cleaved poly (ADP-ribose) polymerase PARP p85 (1:300; Promega, G73411) were used to determine apoptosis induction in cell blocks. Staining was performed in a DAKO Autostainer Link48 (Agilent, Santa Clara, CA, USA) following the manufacturer’s instructions, using 4 μm paraffin sections of cell pellets. All slides were examined by a blind observer under an optic microscope and representative pictures were taken using an Olympus DP73 camera. Positively stained cells were identified using QuPath software v.0.2.3 (Queen’s University, Belfast, Northern Ireland, UK) [21] at 400× magnification and compared to control samples.

### 2.2. Animals

Five-week-old female Swiss nude (Crl:NU(Ico)-*Foxn1^nu^*) and NSG (NOD.Cg-*Prkdc^scid^ Il2rg^tm1Wjl^/SzJ*) mice (Charles River, Écully, France) used for the experiments were maintained under specific pathogen-free conditions, being fed with irradiated food (14% protein, 4% fat, Teklad Global diet, ENVIGO) and reverse osmosis autoclaved water ad libitum. Their light/dark cycle was 12 h and they were housed in individually ventilated cage units (Sealsafe Plus GM500, Techniplast, Buguggiate, Italy) in groups of five. All procedures were performed using sterile material in a type II biosafety cabinet (BIO-II-A/P, Telstar, Terrassa, Spain). During surgery, eye dryness was avoided using a saline drip and body temperature was maintained using a heating blanket under a drape.

The correct implantation in vivo of EC cells, either in the subcutaneous or the orthotopic model, was assessed using firefly D-luciferin (Perkin Elmer, Waltham, MA, USA), a substrate for luciferase that is expressed in transduced AN3CA cells. Mice were injected intraperitoneally with 2.25 mg/mouse, and anesthetized with 3% isoflurane in oxygen to capture the intensity of the emitted bioluminescence (BLI) 5 min after injection. Images were analyzed using Living Image v.4.7.3. software (Perkin Elmer, Waltham, MA, USA).

The endpoint criteria included 10–20% body weight loss, signs of pain or distress, such as abnormal postures or development of ulcers, alopecia, ruffled fur, abnormal breathing, abnormal activity, coma, ataxia or tremors. All mice were euthanized by cervical dislocation at the experimental endpoint.

All the in vivo procedures were approved by the Hospital de la Santa Creu i de Sant Pau Animal Ethics Committee and Generalitat de Catalunya (FUE-2018-00819447, 24 April 2019) and performed according to European Council directives (CEA-OH/9721/2).

#### 2.2.1. Antitumor Effect of Nanotoxins in a Subcutaneous CXCR4^+^ EC Model

In total, 40 Swiss nude mice were anesthetized with 2% isofluorane, and the back was swabbed with povidone-iodine. Then, 10^7^ cells were inoculated subcutaneously (SC) in both flanks. Tumor growth was monitored using a caliper and the formula V = width^2^ × length/2, twice a week.

To assess the effect of each nanotoxin on tumor growth and mice survival, animals were recruited when tumors reached 30–70 mm^3^. They were randomized into three groups (*n* = 10/group), receiving either 10 µg T22-DITOX-H6, 10 µg T22-PE24-H6 or carbonate buffer as a control, three times a week for a maximum of 15 doses (Appendix A).

The tumor size endpoint criterium was set at a volume of 900 mm^3^. In case mice did not reach this criterium after the administration of the 15th dose, they were left untreated until the endpoint.

To determine the activation of protein markers of apoptotic cell death, animals were recruited when tumors reached 150–200 mm^3^ to receive a single bolus of either T22-DITOX-H6 (30 μg), T22-PE24-H6 (100 μg) or carbonate buffer (*n* = 2 per group). Mice were euthanized at 24 or 48 h after treatment.

Finally, tumor and non-tumor organs (lung, kidneys, liver, spleen) were collected, fixed in 4% formaldehyde and paraffin-embedded for histopathological or IHC evaluation of metastasis, apoptotic markers and toxicity.

#### 2.2.2. Antimetastatic Effect of T22-DITOX-H6 Using a Highly Metastatic Orthotopic CXCR4^+^ EC Model

In total, 22 NSG mice were anesthetized with 100 mg/kg ketamine (Ketolar, Pfizer, New York, NY, USA) and 10 mg/kg xylacyn (Rompun, Bayer, Barmen, Germany). Then, 10^6^ CXCR4^+^ Luciferase^+^ AN3CA cells resuspended in 25 uL of culture medium were inoculated through the myometrium into the endometrial cavity using a 29G Hamilton syringe (Microliter Serie 800, Hamilton, Reno, NV, USA), according to the procedure described in [16].

Three days after implantation, cell engraftment and the bioluminescence emission were assessed using IVIS Spectrum 200. Mice were then separated into two groups (*n* = 10/group) to receive either 5 µg T22-DITOX-H6 or carbonate buffer as control, three times a week. Whole body BLI was determined twice a week (Appendix A).

All mice were euthanized when at least one of them reached endpoint criteria, such as a high primary tumor or peritoneal carcinomatosis growth or abdominal distension, which happened two days after the 14th dose. All clinically relevant organs (uterus and ovaries, peritoneal carcinomatosis, lung, kidneys, liver, spleen, abdominal lymph nodes) were collected, fixed in 4% formaldehyde and paraffin-embedded for histopathological analysis, IHC evaluation of metastatic foci and possible toxicity of T22-DITOX-H6 on normal organs.

#### 2.2.3. Histological Examination

Paraffin sections of all organs were stained with hematoxylin-eosin (H&E) and analyzed by two independent blind observers to assess the possible toxicity on non-tumor organs.

#### 2.2.4. Immunohistochemistry

Immunohistochemical staining was performed in a DAKO Autostainer Link48 (Agilent) following the manufacturer’s instructions, using 4 μm paraffin sections of cell pellets or organs. Anti-human vimentin (Dako IGA63061-) was used to determine the presence of tumor cells in clinically relevant organs. Anti-active caspase-3 (1:300; BD Pharmingen, 559565) and anti-cleaved poly (ADP-ribose) polymerase PARP p85 (1:300; Promega, G73411) were used to determine apoptosis induction in vivo.

All slides were examined under an optic microscope and representative pictures were taken using an Olympus DP73 camera. Liver sections were processed with cellSens software (Olympus), while lung sections were analyzed with QuPath, both at 200× magnifications.

### 2.3. Statistical Analysis

All in vitro experiments were performed in biological triplicates. Mice number for in vivo experiments was defined in preliminary experiments to measure the interindividual variability among mice in terms of tumor growth and metastatic load. Randomizations of animals in experimental and control groups were performed using a dice. Two mice with undetectable orthotopic cell engraftment assessed by bioluminescence were excluded from the experiment. The investigator was not blinded to group allocation during in vivo manipulation of animals; nevertheless, she was blinded for the histological analysis. All results were expressed as mean ± standard error (SE). Differences between groups were analyzed using either the Kruskal–Wallis test, Student T-test or Mann–Whitney test, after the determination of normality by the Shapiro–Wilk test. Differences were considered statistically significant at *p* ≤ 0.05. Statistical calculations were performed using SPSS software v.23 (IBM, Armonk, NY, USA).

## 3. Results

### 3.1. Potent CXCR4-Dependent Cytotoxic Effect Induced by T22-DITOX-H6 and T22-PE24-H6 in CE Cells In Vitro

CXCR4 expression was validated in three human EC cell lines in previous work. As previously explained, we selected the transduced AN3CA cell line among ARK-2 and HEC-1A to perform the in vitro work because it internalized our CXCR4-targeted nanocarrier selectively, but also because it showed higher aggressiveness in vivo, obtaining models that could be used afterward in the in vivo evaluation of selected nanoparticles screened in the in vitro studies [16]. Nevertheless, we assessed the cytotoxic activity of two nanotoxins and four distinct proapoptotic nanoparticles in both CXCR4^+^ Luciferase^+^ AN3CA and CXCR4^+^ Luciferase^+^ HEC1A cell lines.

Interestingly, T22-PUMA-GFP-H6, T22-BAXPORO-GFP-H6 and T22-BAK-GFP-H6 proapoptotic nanoparticles did not show cytotoxic effect in any of the tested cell lines after cell exposure to 1000 nM nanoparticles for 48 h. Interestingly, T22-PE24-H6 and T22-DITOX-H6 showed a dramatically higher cytotoxic effect on the CXCR4^+^ AN3CA cell line, compared to the CXCR4^+^ HEC1A cell line (Figure 1A). Thus, we continued the in vitro characterization of the nanotoxins effect using the AN3CA cell line.

After decreasing the concentration range of nanotoxins, both T22-DITOX-H6 and T22-PE24-H6 were found to exert a potent anticancer effect in the low nanomolar rank, with IC50 of 1.6 nM and 2.5 nM, respectively. Moreover, none of the tested nanoparticles in the CXCR4^−^ parental AN3CA cell line showed a cytotoxic effect; therefore, their anticancer effect was CXCR4-dependent (Figure 1B,C). In addition, a competition assay was also performed, by blocking the CXCR4 receptor with CXCR4 antagonist AMD3100 in CXCR4^+^ Luciferase^+^ AN3CA cells, by their pretreatment for 1 h with 1000 nM AMD3100 before nanotoxin exposure. We observed that the cytotoxic effect induced by the T22-DITOX-H6 or T22-PE24-H6 was inhibited using cytotoxic concentrations of the nanotoxins (Figure 1D,E). Therefore, we observed that both T22-DITOX-H6 and T22-PE24-H6 induce a high cytotoxic effect in vitro and also that this activity is mediated by the CXCR4 receptor.

### 3.2. T22-DITOX-H6 and T22-PE24-H6-Induced EC Cell Death In Vitro Is Mediated by Apoptosis Induction

Translocation of phosphatidyl serine was assessed by annexin V/PI assay in CXCR4^+^ Luciferase^+^ AN3CA cells exposed to 4 nM T22-DITOX-H6 or T22-PE24-H6. While early apoptosis only could be assessed in T22-DITOX-H6 treated cells at 6 h, late apoptosis can be found at 24 h in cells treated with any of the two nanotoxins (Figure 2A–C).

Moreover, apoptosis markers such as nuclear condensation, according to the results of DAPI staining, and expression of apoptotic markers (cleaved caspase-3 and its cleavage product, and proteolyzed PARP), increase at 6 h after exposure to both nanoparticles, being this effect higher in cells treated with T22-DITOX-H6 (Figure 2D–F). Positive cells for all these markers were determined using QuPath software and statistically significant differences were found when comparing all treated samples at different timepoints, compared to control cells (Figure 2E,F).

Overall, these results show a high cytotoxic effect in vitro for both nanotoxins through the activation of the apoptotic pathway, as supported by the activation of markers of apoptotic induction markers.

### 3.3. Repeated T22-DITOX-H6 or T22-PE24-H6 Treatment Inhibit Tumor Growth and Increases Survival without Systemic Toxicity in a CXCR4^+^ AN3CA Subcutaneous CE Model

Since both nanotoxins showed a high cytotoxic effect in the CXCR4^+^ Luciferase^+^ AN3CA cell line, we used the subcutaneous EC model derived from this cell line to test their anticancer activity in vivo. Thus, mice bearing subcutaneous tumors derived from this cell line were recruited once the tumors had a volume of 30–70 mm^3^, and were randomized to three groups, which received repeated intravenous doses of either buffer, 10 µg T22-DITOX-H6 or 10 µg T22-PE24-H6. Mice were administered up to 15 doses, and all of them were left to survival, being euthanized once the tumors reached 900 mm^3^.

Nanotoxin-treated groups showed a reduction in tumor growth rate as compared to the control group, as well as a longer mice survival, which was statistically significant for both nanotoxins, and these effects were more pronounced in the T22-DITOX-H6-treated group (Figure 3).

We also performed a single dose T22-DITOX-H6 or T22-PE24-H6 intravenous administration experiment to study the mechanism of cell death induction in tumor tissue, comparing the results of experimental and control groups treated and searching for markers of apoptotic induction. Thus, DAPI staining revealed DNA condensation at 24 and 48 h, while caspase-3 activation could be observed 24 h after treatment to increase their intensity level after 48 h, an effect that was followed by the induction of PARP proteolysis (Figure 4). Altogether, these three indicators reveal activation of the apoptotic pathway in tumor tissue similar to what we had already observed in vitro.

In addition, mice kept normal behavior in all compared groups; moreover, the administration of the nanotoxins did not induce weight loss during the treatment period (Figure 5A). Moreover, the potent antitumor effect exerted by T22-DITOX-H6 and T22-PE24-H6 induced in the subcutaneous EC model showed no histopathological alterations in normal tissues (liver, kidney, lung, spleen) at 48 h post-treatment, as assessed in H&E-stained tissue slides. Thus, no systemic off-target toxicity was found regarding the possible alteration of tissue architecture, infiltrating lymphocytes or protein accumulation (Figure 5B). Notably, the lack of on-target toxicity found in the spleen is a remarkable finding, despite the expression of target receptor CXCR4 by leukocytes that accumulate in this organ. Nevertheless, the CXCR4 expression level in the spleen is significantly lower than the CXCR4 expression found in EC tissues.

### 3.4. T22-DITOX-H6 Repeated Administration Reduces Total Tumor Burden as Measured by Bioluminescence in a CXCR4^+^ AN3CA EC Orthotopic Model

In previous work, we generated the CXCR4^+^ Luciferase^+^ AN3CA-derived model, which showed a high metastatic load at all relevant clinical sites. Using this model, we demonstrated that the nanotoxin-treated group significantly reduced the development of metastatic foci antimetastatic effect as compared to buffer-treated mice [16]. The other two EC cell lines did not generate orthotopic models with sufficient metastatic load; thus, being not usable cancer models to test the antimetastatic effect of any drug in EC. Moreover, the CXCR4^+^ HEC1A cells were only moderately sensitive to the tested therapeutic nanoparticles. Therefore, we used the CXCR4^+^ Luciferase^+^ AN3CA cells in the previously described procedure to generate the metastatic model: NSG mice were injected orthotopically into the endometrium with CXCR4^+^ Luciferase^+^ AN3CA cells. Three days after inoculation to allow cell engraftment, mice were randomized in the control or treated groups to be administered intravenously with repeated doses of either 5 µg T22-DITOX-H6 3 days a week or Buffer following the same schedule. 48 h after the 14th administration, 10 mice met endpoint criteria, and both groups were euthanized.

Bioluminescence emission by EC tumor cells allowed the follow-up of their engraftment, development and organ dissemination. T22-DITOX-H6 treated group showed a significant reduction in whole-body bioluminescence emission, which reflects tumor burden, at days 25 and 32 after treatment start (Figure 6).

### 3.5. Potent Inhibition of the Development of Liver and Lung Metastases by T22-DITOX-H6 Repeated Dosage in a CXCR4^+^ AN3CA EC Model

Metastatic foci development in the liver and lung was inhibited by T22-DITOX-H6 treatment, as measured by human vimentin IHC on tissue slides. We found that 33.3% of nanotoxin-treated mice were free of liver and lung metastasis. Interestingly, a dramatic decrease was found in EC metastatic foci number and foci area in the liver (Figure 7A–C; Appendix A). Regarding lung metastasis, there was a significant decrease in the percentage of mice showing tumor cells in lung, from 100% in the control group to 55.56% in the T22-DITOX-H6-treated group. In addition, nanotoxin-treated mice showing had significantly lower EC cancer cell dissemination in lung tissue than the control group (Figure 7D–F; Appendix A).

### 3.6. Inhibition of Lymph Node Metastases and Carcinomatotic Foci and Metastases after T22-DITOX-H6 Repeated Dosage in the Disseminated EC Model in the Absence of Systemic Toxicity

The development of both lymph node metastases and peritoneal implants was evaluated, as well as the possible toxic effect on non-tumor tissues after repeated doses of T22-DITOX-H6.

Even though all mice showed macroscopic peritoneal implants, significant differences were found in control and treated animals (Figure 8A–C; Appendix A). T22-DITOX-H6 significantly inhibited their growth regarding both the number and total weight of the peritoneal metastases. Nevertheless, differences in the percentage of mice in each group developing lymph nodes metastasis did not reach significance (Figure 8D,E).

Moreover, T22-DITOX-H6 did not induce any off-target toxicity in normal tissues, in spite of the maintenance of the body weight in mice belonging to the nanotoxin-treated group while the body weight in control mice increased over time (Figure 9A). To explain this statement, first, we observed a lack of toxicity in non-target organs such as kidney, liver, lung and spleen. Thus, all these organs showed normal appearance after ex vivo inspection; moreover, no histological alterations were found in H&E-stained tissue sections of each of these organs, which includes the observation of normal architecture in the hepatic parenchyma, and lack of architecture alteration or protein accumulation in kidney, lung or spleen (Figure 9B). Secondly, the higher body weight registered in buffer-treated mice, as compared to nanotoxin-treated animals (Figure 6A), was due to the dramatic growth of the metastatic growth in the peritoneum (Figure 8C) that increased the whole-body weight of the animal. This increase in body weight was not observed in nanotoxin-treated mice because treatment potently inhibits the growth of peritoneal metastases; therefore, maintaining the normal weight of the adult mice (Figure 6A and Figure 8C).

## 4. Discussion

Women bearing EC at an advanced stage or at high risk of developing locoregional or distal metastasis lack effective treatments, and currently approved drugs lead to poor outcomes and severe adverse effects for EC patients.

We, here, first demonstrated in vitro that our nanotoxins exert a potent and CXCR4-dependent anticancer effect, in CXCR4^+^ AN3CA cells (IC_50_ = low nanomolar range), as compared to a moderate effect on HEC1A cells showing lower CXCR4 membrane expression. Moreover, both T22-PE24-H6 and T22-DITOX-H6 nanotoxins induced apoptosis, as detected by Annexin V expression and DNA condensation in apoptotic bodies. Interestingly, T22-DITOX-H6 shows higher early apoptosis compared to T22-PE24-H6, a result that is consistent with the faster DITOX path to reach the cytosol, opening a pore in the endosome, whereas PE24 needs to translocate from the endosome to Golgi and endoplasmic reticulum prior to reaching the cytoplasm. Finally, both toxins trigger cell death by blocking protein synthesis through eEF-2 inhibition [12,22].

Next, we used two previously developed models by our group, derived from CXCR4^+^ EC AN3CA cells, a subcutaneous (SC) and a highly metastatic EC model, which allows the detection of single EC cells invading distant organs because of their expression of human Vimentin [16]. Thus, repeated intravenous administration of each nanotoxin in the SC EC model showed a highly significant antineoplastic effect, also mediated by apoptosis with DNA condensation and Caspase-3 and PARP cleavage. Moreover, this treatment increased survival compared to control mice and lacked on-target or off-target systemic toxicity; therefore, implying a wide therapeutic window in terms of harm–benefit balance for both nanotoxins.

In addition, repeated intravenous T22-DITOX-H6 administration in the CXCR4^+^ EC AN3CA orthotopic model induced a potent antimetastatic effect by displaying a reduction in primary tumor growth, and a blockage of metastasis development, again without systemic toxicity. Furthermore, peritoneal metastases were reduced by 50%, while lymph node metastases were diminished by 60%; however, the study of their significance was limited by the direct contact and tissue merging between lymph nodes and peritoneal metastases. On the other hand, T22-DITOX-H6 shrinks 5-fold the number of liver metastases and 10-fold their size compared to control mice. Finally, nanotoxin treatment significantly reduced 8-fold the total area of the lung invaded by metastatic cancer cells. Thus, it is highly likely that this potent antimetastatic effect occurs through the elimination of CXCR4^+^ cells while they migrate, invade or grow at the colonized organs.

The relevance of our original approach relies on its high therapeutic potency in the absence of systemic toxicity. In contrast, several immunotoxins, in which antibodies targeting HER-2 are bound to PE24 or DITOX, are being tested in early clinical trials that include EC patients; however, they still develop severe side effects [23]. Moreover, in 2004, a molecular therapy targeting HER-2, incorporating active caspase domains and the cytotoxic domain of P. aeruginosa (the same employed in T22-PE24-H6) was successful in preclinical experiments but, as far as we know, it did not reach clinical trials [24,25,26]. Recently, antitumor the effect of angubindin-1 (derived from *C. perfingens* iota toxin) has been evaluated in vitro, but no preclinical in vivo or clinical results have been published to date [27,28].

On the other hand, and as previously explained, immunotoxins have been approved only for leukemia or lymphoma, while they have not been effective enough on solid tumors, due to limitations such as low accessibility to tumor cells and their toxicity and immunogenicity [11]. In contrast, several antibody-drug conjugates (ADCs) are currently being tested in clinical trials for EC to target the Folate receptor-α Tissue factor, HER-2 or Trop-2, which are conjugated to microtubule inhibitors (DM-4, MMAE) or genotoxic agents (deruxetan, SN-38 or Duocarmycin) [8]. It is relevant to indicate that all of these approaches clearly differ regarding the target cancer cells since our nanotoxins target the CXCR4 receptor.

It is equally important to remark that, instead of delivering microtubule inhibitors or genotoxic agents to the cytosol in the case of ADCs, the T22-DITOX-H6 nanotoxin delivers the exotoxin domain to the cytosol to activate an alternative mechanism of action that consists of the inhibition of eEF-2. This molecular inhibition blocks protein translation to induce cancer cell death not only in dividing tumor cells but also in cancer cells in the Go phase of the cell cycle. This mechanism, never studied in EC patients, could have a clinical impact because of its capacity to prevent metastasis development. Furthermore, no immunotoxins are being tested in EC patients, while the ADCs being tested for EC are still in early clinical phases (I or II), and there are no significant results so far in EC to determine whether or not they are showing activity. However, severe adverse effects that limit the dosage have been reported in these trials [29,30,31], which are consistent with the reports on ADCs that deliver microtubule inhibitors or genotoxic agents and also the fact that only ~0.1% of the ADC administered drug reaches cancer tissues [32]. All these limitations seem to be overcome by the use of our multivalent display of multiple ligands for the CXCR4 receptor in our nanotoxin, which makes for a highly selective delivery of the bacterial cytotoxin domain to CXCR4^+^ EC cells. Therefore, their highly selective delivery is likely to obtain higher cancer tissue uptake and achieve an effective control of tumor growth or dissemination, without on-target or off-target toxicity in normal organs.

In addition, we believe that our nanotoxins could be combined with other drugs already explored for EC therapy, which would include a wider range of candidate patients for treatment. They are PI3K/Akt inhibitors (perifosin, BKM120, temsirolimus or everolimus) or immune checkpoint inhibitors (pembrolizumab, dostarlimab), which have been FDA-approved for microsatellite instability-high and mismatch repair deficient patients; thus, their combination could enhance the current success rate by lowering their dosages and reducing adverse effects. Of note, for the in vivo assessment of nanotoxins, we have used CXCR4^+^ mice models derived from the AN3CA cell line, which has described both alterations of the PI3K/Akt pathway and high microsatellite instability [33]. Furthermore, this is a novel and highly metastatic model that overcomes the low metastatic rates shown in previous models; previously developed anticancer drugs that are currently applied were never tested in EC disseminated models. One major limitation of immunotoxin development is its severe adverse effects. They could be due, in part, to the targeting antibodies that the immunotoxins carry, which display only two binding sites, whereas our nanotoxins have a significantly higher multivalency since they are estimated to reach as far as eleven T22-ligands for the target CXCR4 receptor. On the other hand, being aware of the immunogenicity associated with immunotoxicity treatment, we plan to carry out further in vivo experiments using immunocompetent mice strains to better evaluate if the nanotoxins induce the activation of the immune system to increase the antitumor effect and whether they trigger antibodies against the toxin that could neutralize their effectiveness, before the initiation of their clinical translation. In summary, and based on all described results, we propose that the nanotoxin T22-DITOX-H6, and also most likely T22-PE24-H6, emerge as promising therapeutic candidates for their further preclinical and regulatory development to be performed before entering clinical trials in advanced CXCR4^+^ EC patients, which could improve the low selectivity of current treatments for this aggressive illness.

## 5. Conclusions

T22-DITOX-H6 and T22-PE2-H6 nanotoxins show a highly CXCR4-dependent cytotoxic effect through the activation of the apoptotic pathway in EC cells. Both nanotoxins, but especially T22-DITOX-H6 at a higher degree, reduce tumor growth and prolong animal survival after their repeated administration in a subcutaneous CXCR4^+^ EC model. Moreover, the intravenous administration of T22-DITOX-H6, following a repeated dose schedule, in a highly metastatic CXCR4^+^ EC mouse model, previously developed by our group, yielded a dramatic decrease in overall tumor burden and potent blockade of dissemination of peritoneal, liver and lung metastasis, in the absence of off-target or on-target systemic toxicity. These results indicate that T22-DITOX-H6 shows a potent preclinical anticancer effect, thus being a potential treatment for high-risk EC patients. Remarkably, CXCR4 is overexpressed in a majority of EC patients, while advanced-staged EC patients lack effective therapies, making this nanotoxin a promising therapeutic alternative for further preclinical and clinical development.

## Figures and Tables

**Figure 1 cancers-15-00085-f001:**
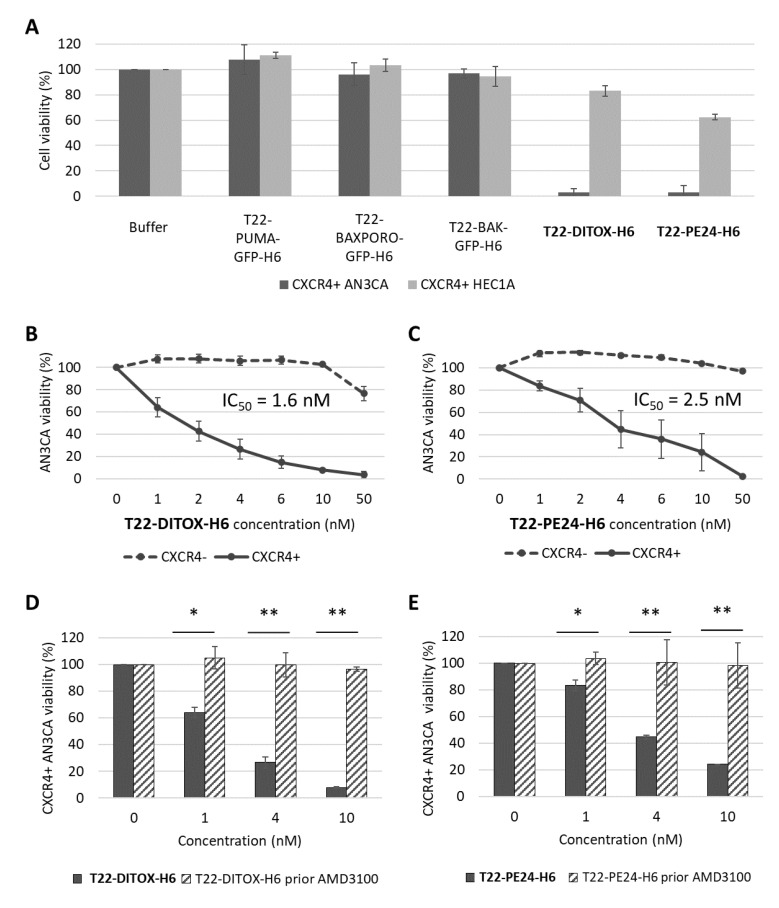
The CXCR4-dependent cytotoxic effect induced by T22-DITOX-H6 and T22-PE24-H6, at a low nanomolar range, in human endometrial cancer cells in vitro. (**A**) Screening of therapeutic nanoparticles targeted to the CXCR4 receptor, at 1 µM protein concentration, in the CXCR4^+^ AN3CA and CXCR4^+^ HEC1A cell lines. T22-DITOX-H6 and T22-PE24-H6 show a higher cytotoxic effect on CXCR4^+^ AN3CA cells than the rest of pro-apoptotic nanoparticles. (**B**,**C**) Dose–response curve expressed as percent of remaining viable CXCR4^+^ and CXCR4^-^ AN3CA cells after exposure to T22-DITOX-H6 (**B**) or T22-PE24-H6 (**C**) at the 1–50 nM range. (**D**,**E**) Competition assays, measuring the percent of viable CXCR4^+^ AN3CA cells after exposure to T22-DITOX-H6 (**D**) or T22-DITOX-H6 (**E**) without or with a prior exposure to the CXCR4 antagonist AMD3100. * *p* < 0.005; ** *p* = 0.001 (mean ± SEM; Mann–Whitney).

**Figure 2 cancers-15-00085-f002:**
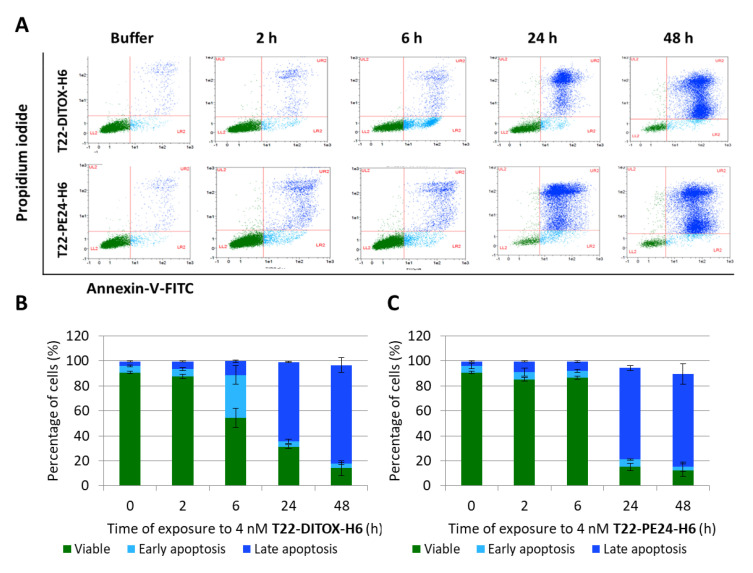
T22-DITOX-H6 and T22-PE24-H6-induced cell death in vitro in CXCR4^+^ AN3CA endometrial cancer cells is mediated by apoptosis. (**A**) Representative dot-plot and quantitation of viable cells (green), at early (light blue) or late (navy blue) apoptosis using Annexin V-FITC/PI assay, after their exposure for 2, 6, 24 and 48 h to 4 nM of T22-DITOX-H6 (**B**) or T22-PE24-H6 (**C**), expressed as mean ± SEM. (**D**) Representative images of DAPI or immunohistochemistry staining of the apoptosis mediators, active cleaved caspase-3 and cleaved PARP. Experiments were performed using CXCR4^+^ AN3CA cell pellets after 6, 12, 24 or 48 h exposure to 4 nM of T22-DITOX-H6 or T22-PE24-H6. Bar: 50 µm. (**E**,**F**) Quantification of positive cell percentage, compared to control, regarding the induction of apoptotic bodies (DAPI staining) and cleavage of caspase-3 (act. Casp-3) and PARP (pPARP) (Mann–Whitney; *p* < 0.008 in all cases).

**Figure 3 cancers-15-00085-f003:**
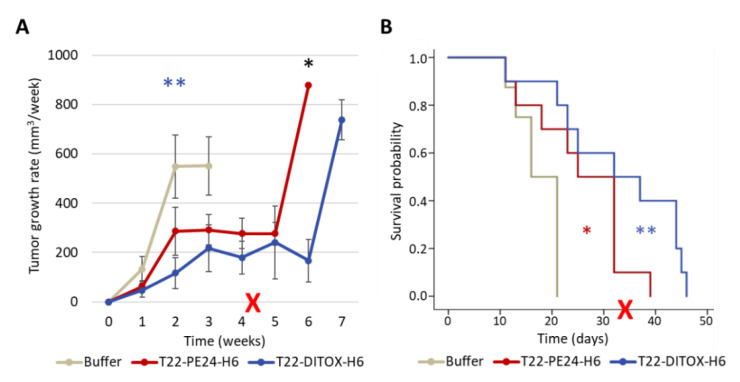
Repeated dose of T22-DITOX-H6 or T22-PE24-H6 treatment inhibit tumor growth and increases survival without systemic toxicity in a CXCR4^+^ AN3CA subcutaneous EC model. Mice were treated with either buffer, 10 µg T22-DITOX-H6 or 10 µg T22-PE24-H6, 3 times a week, until a maximum of 15 doses. (**A**) Weekly tumor growth rate over time measured in all three groups ** *p* = 0.005; * *p* = 0.034 (mean ± SEM; T-test). (**B**) Mouse survival time registered for the buffer and nanotoxin-treated groups. * *p* = 0.017; ** *p* = 0.002 (mean ± SEM; Log-Rank test). The red cross indicates the end of treatment doses.

**Figure 4 cancers-15-00085-f004:**
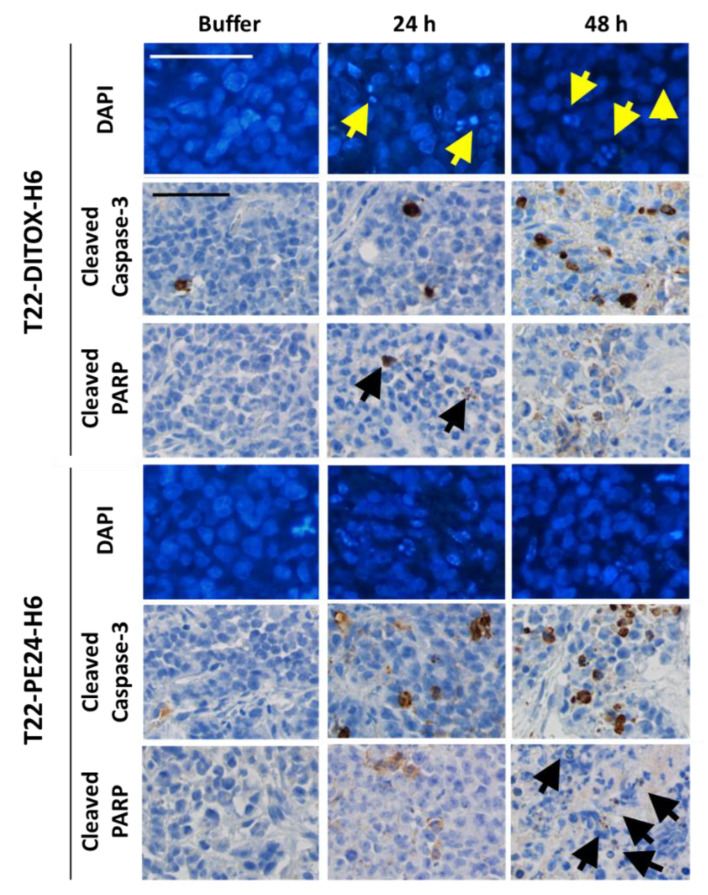
T22-DITOX-H6 and T22-PE24-H6 nanotoxin treatment activates apoptosis in vivo in a subcutaneous tumor generated in the CXCR4^+^ AN3CA EC model. Mice were treated with a single bolus of either buffer, 100 µg T22-DITOX-H6 or 30 µg T22-PE24-H6. Tumor sections were stained with fluorescent DAPI, showing DNA condensation in apoptotic bodies, and IHC staining for markers of cleaved caspase-3 and proteolyzed PARP. Arrows indicate positive cells. Bar: 50 μm.

**Figure 5 cancers-15-00085-f005:**
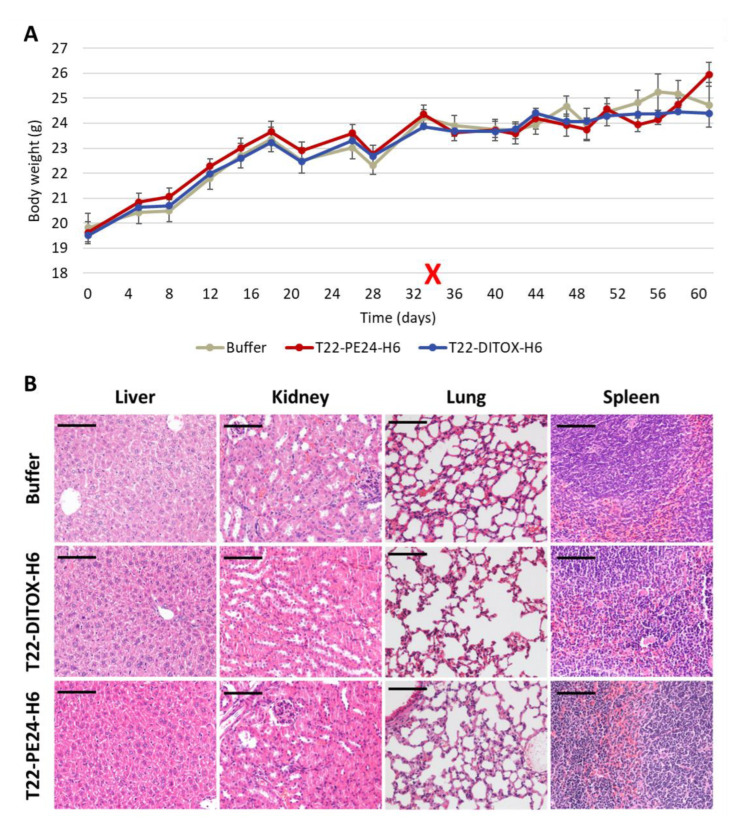
T22-DITOX-H6 and T22-PE24-H6 did not induce systemic toxicity in the CXCR4^+^ AN3CA subcutaneous EC model. (**A**) Body weight evolution over time did not show differences between control and nanotoxin-treated mice. The red cross indicates the end of treatment administration. (**B**) Hematoxylin-eosin staining of non-tumor organs 48 h after the last nanotoxin administration showed normal tissue architecture and, therefore, a lack of histological alterations and lack of protein deposition. Bar: 100 µm.

**Figure 6 cancers-15-00085-f006:**
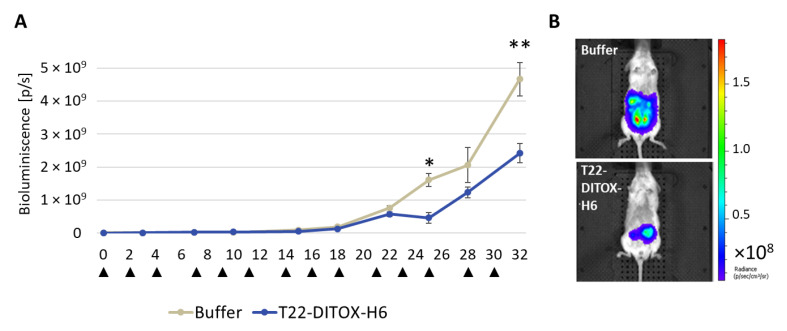
T22-DITOX-H6 repeated dosage reduces whole-body bioluminescence emission in an orthotopic CXCR4^+^ AN3CA EC model. (**A**) Total bioluminescence intensity (BLI) registered over time in mice treated with buffer or T22-DITOX-H6 (*n* = 9 animals per group). Data are reported as mean ± SEM. Asterisks indicate significant differences in emitted BLI between the compared groups using the Mann–Whitney U test (* *p* < 0.05; ** *p* < 0.01). Arrows indicate treatment administration. (**B**) Representative images of the control and nanotoxin-treated mice as registered by IVIS Spectrum at day 32.

**Figure 7 cancers-15-00085-f007:**
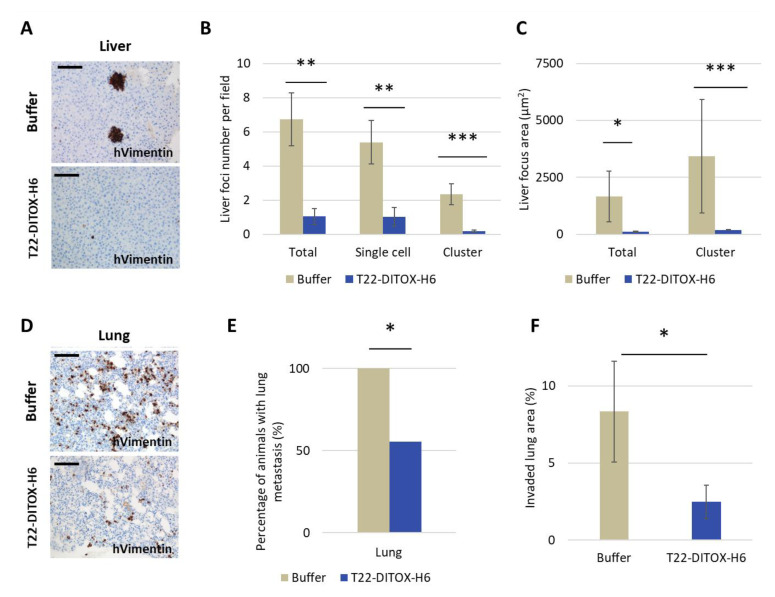
Antimetastatic effect measured by IHC detection of human cancer cells in liver and lung tissues after T22-DITOX-H6 repeated dose treatment in a CXCR4^+^ AN3CA endometrial cancer model. Mice were treated with either buffer or 5 μg T22-DITOX-H6 for a total of 14 doses. (**A**) Representative images of immunohistochemical (IHC) staining of human vimentin registered in liver tissue for each group. (**B**) Quantification of liver metastatic foci number, including total foci number and the number of cell foci or clustered cell foci (** *p* ≤ 0,005; *** *p* = 0,000; mean ± SEM; Mann–Whitney U test). (**C**) Comparison of metastatic foci area in liver occupied by all foci or only for clustered cell foci (* *p* = 0.023; *** *p* = 0.001; mean ± SEM; Mann–Whitney U test). (**D**) Representative images of IHC staining for human vimentin of lungs registered in control and treated mice. (**E**) Comparison of the percent mice displaying positive lung metastasis between control and nanotoxin-treated mice (* *p* = 0.041; Fisher test). (**F**) Comparison of the lung tissue area invaded by human tumor cells between control and nanotoxin-treated mice (* *p* = 0.038; mean ± SEM; Mann–Whitney U test). Bars: 100 μm.

**Figure 8 cancers-15-00085-f008:**
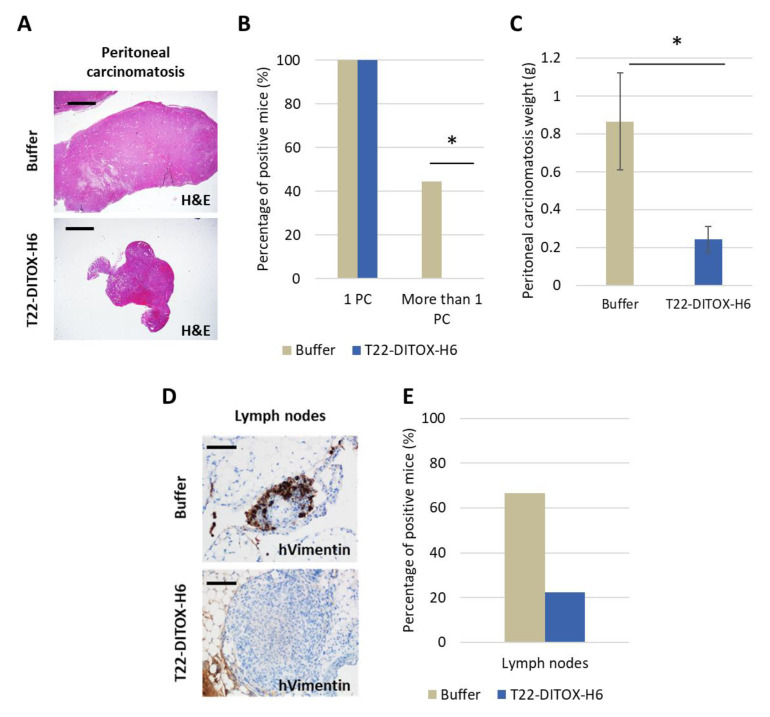
Antimetastatic effect in peritoneal and lymph node tissue after T22-DITOX-H6 repeated administration in the disseminated CXCR4^+^ AN3CA endometrial model. Mice were treated with either buffer or 5 ug T22-DITOX-H6 for a total of 14 doses. (**A**) Representative images of hematoxylin-eosin staining of peritoneal carcinomatosis in buffer or nanotoxin-treated mice (bar: 2 mm). (**B**) Percentage of control or nanotoxin-treated mice bearing one or more than one peritoneal carcinomatotic implant (* *p* = 0.041; Fisher test). (**C**) Comparison of the total weight of peritoneal carcinomatotic foci registered in control and nanotoxin-treated mice (* *p* = 0.036; Mann–Whitney U test). (**D**) Representative images of lymph node metastases in control and nanotoxin-treated mice by IHC detection of human vimentin (bar: 100 μm). (**E**) Percentage of control and nanotoxin-treated mice with positive lymph node metastases at endpoint.

**Figure 9 cancers-15-00085-f009:**
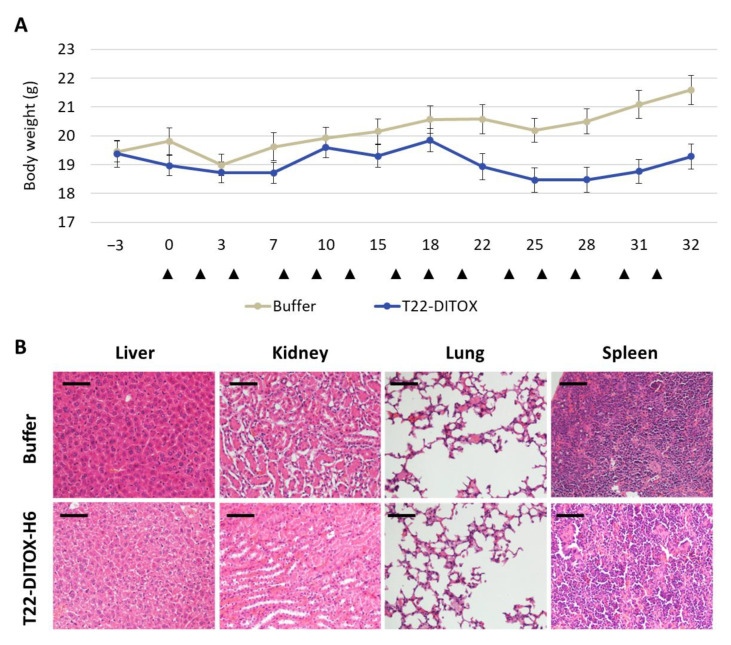
Lack of systemic toxicity after T22-DITOX-H6 repeated administration in the disseminated CXCR4^+^ AN3CA endometrial model. (**A**) Body weight evolution over time. Black arrows indicate treatment administration (**B**) Histological analysis of hematoxylin-eosin-stained sections of non-tumor organs (liver, kidney, lung and spleen) in mice treated with buffer or T22-DITOX-H6 (bar: 100 μm). Notice the lack of histopathological alterations in all studied tissues.

## Data Availability

The datasets generated during and/or analyzed during the current study, as well as additional information and data, are available from the corresponding author upon reasonable request.

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
