# Peer review of "Potent Anticancer Activity of CXCR4-Targeted Nanostructured Toxins in Aggressive Endometrial Cancer Models"

_cancers, 2022, doi:10.3390/cancers15010085_

Round 1

Reviewer 1 Report

The study by Medina-Gutiérrez et. al. studied the efficacy of two nanotoxins, namely T22-DITOX-H6 and T22-PE2-H6, on CXCR4-expressing endometrial cancerous cells. The short and descriptive study provides minor novelty. They show that both nanotoxins, in vitro and in vivo, exerted a potent antitumor effect via apoptosis, and extended the survival of nanotoxin-treated mice without inducing any off-target toxicity. The study is interesting and provides novel inputs. However, some concerns regarding the study's technical conductance must first be addressed.

Comments:

1-     Additional viability assay is required. Assays like colony formation or BrdU are important for the proliferation effect.

2-     Why do different cell lines differ in the internalization of the nanocarrier? In this case, how do the authors reckon the uptake efficiency by human cancer subjects?

3-     Figure 5, panels showing the lungs of the nano toxins-treated mice, they don’t look normal to me, at least not as in buffer-treated lungs.

4-     Data on body weight change and hematological parameters are needed to evaluate the cytotoxicity of the treatments

5-     Why specifically vimentin was used as a malignant marker? It is only positive for cancerous cells undergoing epithelial-to-mesenchymal transition

6-     The Discussion section is too long and does not focus. It mostly repeats the introduction and the results. It should first summarize the major findings in the study, followed by discussing the differences in the study and the literature, the limitation of this study, and the conclusion of this study and future work.

Author Response

The study by Medina-Gutiérrez et. al. studied the efficacy of two nanotoxins, namely T22-DITOX-H6 and T22-PE2-H6, on CXCR4-expressing endometrial cancerous cells. The short and descriptive study provides minor novelty. They show that both nanotoxins, in vitro and in vivo, exerted a potent antitumor effect via apoptosis, and extended the survival of nanotoxin-treated mice without inducing any off-target toxicity. The study is interesting and provides novel inputs. However, some concerns regarding the study's technical conductance must first be addressed.

Comments:

POINT 1-     Additional viability assay is required. Assays like colony formation or BrdU are important for the proliferation effect.

Response 1: While we did not assess the possible block of cell proliferation in this work, we believe that we provide sufficient mechanistic studies on death induction through apoptosis in EC cells, both in vitro and in vivo, to explain the anticancer activity found by each nanotoxin treatment. Of course, we recognize the importance of the blockade of cell proliferation by anticancer drugs; however, we believe that the translation of this proliferative block into anticancer effect requires the subsequent trigger of cell death. On this basis, our in vitro studies demonstrated a potent reduction of EC cell viability, using the XTT metabolic assay, and the detection of annexin-V by flow cytometry that indicate phosphatidylserine translocation to the external membrane, a marker of apoptotic induction. Moreover, in the in vivo model, we found expression of Caspase-3 and cleavage PARP in tumor tissues after T22-DITOX-H6 or T22-PE24-H6 therapy, both being also markers that mediate the induction of apoptosis. In addition, the described results are also consistent with the capacity of both nanotoxins to block protein translation to trigger cancer cell death.

POINT 2-        Why do different cell lines differ in the internalization of the nanocarrier? In this case, how do the authors reckon the uptake efficiency by human cancer subjects?

Response 2: In previous in EC cells in vitro, and in in vivo biodistribution assays using EC models (Medina-Gutiérrez et al., 2022; https://www.mdpi.com/2227-9059/10/7/1680), the internalization of our fluorescent T22-GFP-H6 nanocarrier (which displays the exact nanostructure as the nanotoxins) is completely dependent on CXCR4 expression in the EC cell membrane. Thus, the i.v. bolus administration of T22-GFP-H6 in SC models of different cancer types consistently shows around 60-80% uptake of the injected dose in tumor tissue (Medina-Gutiérrez et al., 2022; doi.org/10.3390/biomedicines10071680; Rioja-Blanco et al., 2022; doi: 10.1016/j.apsb.2021.09.030; Falgàs A et al. 2020, doi: 10.3324/haematol.2018.211490). 

Consistently, EC cell lines with high CXCR4 membrane expression (i.e AN3CA cell line) display higher nanoparticle CXC4-targeted nanotransporter internalization than the one registered in the EC HEC1A cells that showed a mixed pattern of cytoplasmic and lower membrane CXCR4 expression. Similarly, CXCR4-targeted nanotoxins induce higher level of cytotoxicity in CXCR4+ AN3CA cell line than in CXCR4+ HEC1A cells (supplementary material in this publication).

In this regard, CXCR4 overexpression in EC patients is mostly found in cancer cell membrane, therefore, AN3CA cells better represent the clinical situation, being this cell line a more suitable method to test the anticancer effect of the CXCR4-targeted nanotoxins and its CXCR4-dependence; and therefore, being better models for the preclinical evaluation of CXCR4-targeted drugs. Moreover, and regarding clinical translation, we are proposing that clinical trials recruit only patients with high level of membrane CXCR4 in EC primary tumor biopsies to be treated with these nanotoxins.

POINT 3-        Figure 5, panels showing the lungs of the nano toxins-treated mice, they don’t look normal to me, at least not as in buffer-treated lungs.

Response 3: Thank you for your comment. Based on your suggestion, we asked our pathologist to check the image of the lung sample that represents the buffer treated animals. Despite the pathologist saw no histological alteration in the displayed lung section, we decided to perform again a sample assessment of the lung architecture in randomly chosen sections of different mice belonging to the control or experimental groups. Again, the histology in buffer-treated lung tissue was indistinguishable from this found in lung sections of nanotoxin-treated mice. Thus, we believe that the selected histology correctly represents lung tissue in the buffer-treated group; thus, we decided to keep this image in Figure 5. However, if you prefer, we will change the image by another area of the lung tissue with higher similarity to those found in nanotoxin-treated groups.

POINT 4-        Data on body weight change and hematological parameters are needed to evaluate the cytotoxicity of the treatments

Response 4: Following this Reviewer request, we have now included a new panel in Figure 5 showing the lack of significant body weight changes in nanotoxin-treated mice during the experiment in the subcutaneous EC model.

Also, for the orthotopic EC model we present a panel (Figure 9.A) showing the absence of a significant change in body weight (<10% of mice weight at the beginning of the experiment) in nanotoxin-treated mice along time. Besides both, control and treated groups, show similar weight evolution, the apparent reduction of weight in T22-DITOX-H6-treated mice is not due to systemic toxicity, since no histological alteration has been observed in normal tissues. Instead, it could most likely be due to the block of primary tumor and carcinomatotic foci, which maintain the animal body weight. In contrast, body weight of buffer-treated animals dramatically increases because of the development of large peritoneal metastases, as shown in Figure 8.C. This is also supported by T22-DITOX-H6 effective inhibition of whole cancer burden, as measured by total body bioluminescent emitted by EC cells at the different sites (reported in Figure 6).

On the other hand, the study of possible hematological toxicity was not performed in this work because of our previous and exhaustive studies, reported in two recent publications by our group. Since both these articles describe lack of alteration in mice treated with the same nanotoxin dosage as the one here used work (Falgàs, 2022 (https://pubmed.ncbi.nlm.nih.gov/35421785/); Rioja-Blanco, 2022 (https://www.ncbi.nlm.nih.gov/pmc/articles/PMC8815235/), we did not consider it necessary in the present. In these two publications we found no changes in white blood cell count, including neutrophils, lymphocytes, monocytes, eosinophils, or basophils, and no changes in red blood cell count, platelets, hemoglobin, hematocrit, mean cell volume, mean corpuscular hemoglobin, mean corpuscular hemoglobin concentration, red blood cell distribution width, platelet count, mean platelet volume, platelet distribution width or plateletcrit.

POINT 5-        Why specifically vimentin was used as a malignant marker? It is only positive for cancerous cells undergoing epithelial-to-mesenchymal transition

Response 5: A large percent of advanced endometrial carcinomas display markers of epithelial-to-mesenchymal transition (EMT) in cancer cells, which includes Vimentin expression. Moreover, positive EMT is a significant predictor of shorter progression-free survival and shorter overall survival (Yoshito Terai Y et al. Cancer Biology & Therapy 20013, 14:1, 13–19; doi: 10.4161/cbt.22625p). Thus, we were lucky to find the CXCR4+ EC AN3CA cell line, which, besides of having underwent an EMT, it displays high expression of human vimentin in vitro and also in vivo, that is, in the cancer models derived from this cell line. This fact allows the easy detection of human EC cells in mouse tissue, using immunohistochemistry, that is highly sensitive and specific method especially useful in the detection of single or small microscopic groups of EC cells invading distant organs. Its high specificity derives also from the fact that the anti-hVimentin antibody by Dako does not cross-react with mouse vimentin.

Thus, our methodological approach overcomes the insufficient sensitivity regarding metastasis detection, which is a major drawback to follow metastatic spread in EC models or in their use in the development of novel antimetastatic therapies for EC. Most frequently employed methods, namely bioluminescence and hematoxylin-eosin staining, do not allow the detection of single cell invasion or colonization of clinically relevant organs, which we considered a loss of relevant information on cancer dissemination to organs divers from the endometrium.

POINT 6-      The Discussion section is too long and does not focus. It mostly repeats the introduction and the results. It should first summarize the major findings in the study, followed by discussing the differences in the study and the literature, the limitation of this study, and the conclusion of this study and future work.

Response 6: We have shortened the length of the discussion in the manuscript by making a statement on the importance of the development of novel therapies for advanced EC, followed by a synthetic description of the main results obtained, to concentrate in discussing the results that add or change the current state of the art on EC therapy and future work as well as the limitations of the study. 

Reviewer 2 Report

The article represents a significant and promising study. So , I recommend to publish.

Author Response

Response: Thank you very much for your positive comments and evaluation.

Reviewer 3 Report

The incidence of endometrial cancer is increasing annually, and the trend of younger age is obvious.This study generated novel CXCR4-targeted nanotoxins, T22-DITOX-H6 and T22-PE24-H6 and evaluated their antitumor and anti-metastatic effect of the T22-DITOX-H6 and T22-PE24-H6 in vivo and in vitro. It has some clinical significance. Secondly, the design of this study is reasonable, and the conclusions are reliable. It is suggested that publication could be considered after some revisions. However, there are some shortcomings in the article: 1. The labeling of some references is somewhat inconsistent, such as lines 61-62 on the second page, 92 lines on the third page, etc.; 2. The language part of the article needs further revision.

Author Response

The incidence of endometrial cancer is increasing annually, and the trend of younger age is obvious.This study generated novel CXCR4-targeted nanotoxins, T22-DITOX-H6 and T22-PE24-H6 and evaluated their antitumor and anti-metastatic effect of the T22-DITOX-H6 and T22-PE24-H6 in vivo and in vitro. It has some clinical significance. Secondly, the design of this study is reasonable, and the conclusions are reliable. It is suggested that publication could be considered after some revisions. However, there are some shortcomings in the article: 1. The labeling of some references is somewhat inconsistent, such as lines 61-62 on the second page, 92 lines on the third page, etc.; 2. The language part of the article needs further revision.

Response: Thank you for your comments. Reference inconsistencies have been checked thoroughly, and they have been corrected. We have also revised the language to make the sentences and paragraphs more simple and direct, and have also amended some spelling mistakes.